# Ruling out acute coronary syndrome in primary care with a clinical decision rule and a capillary, high-sensitive troponin I point of care test: study protocol of a diagnostic RCT in the Netherlands (POB HELP)

Simone van den Bulk [1], Annelieke H J Petrus [1], Robert T A Willemsen [2], Mark J Boogers [3], Joan G Meeder [4], Braim M Rahel [4], M Elske van den Akker-van Marle [5], Mattijs E Numans [1], Geert-Jan Dinant [2], Tobias N Bonten [1]

**To cite:** van den Bulk S, Petrus AHJ, Willemsen RTA, *et al.* Ruling out acute coronary syndrome in primary care with a clinical decision rule and a capillary, high-sensitive troponin I point of care test: study protocol of a diagnostic RCT in the Netherlands (POB HELP). *BMJ Open* 2023;13:e071822. doi:10.1136/bmjopen-2023-071822

For numbered affiliations see end of article.

**Correspondence to**
Simone van den Bulk;
s.van_den_bulk@lumc.nl

## ABSTRACT

**Introduction** Chest pain is a common reason for consultation in primary care. To rule out acute coronary syndrome (ACS), general practitioners (GP) refer 40%–70% of patients with chest pain to the emergency department (ED). Only 10%–20% of those referred, are diagnosed with ACS. A clinical decision rule, including a high-sensitive cardiac troponin-I point-of-care test (hs-cTnI-POCT), may safely rule out ACS in primary care. Being able to safely rule out ACS at the GP level reduces referrals and thereby alleviates the burden on the ED. Moreover, prompt feedback to the patients may reduce anxiety and stress.

**Methods and analysis** The POB HELP study is a clustered randomised controlled diagnostic trial investigating the (cost-)effectiveness and diagnostic accuracy of a primary care decision rule for acute chest pain, consisting of the Marburg Heart Score combined with a hs-cTnI-POCT (limit of detection 1.6 ng/L, 99th percentile 23 ng/L, cut-off value between negative and positive used in this study 3.8 ng/L). General practices are 2:1 randomised to the intervention group (clinical decision rule) or control group (regular care). In total 1500 patients with acute chest pain are planned to be included by GPs in three regions in The Netherlands. Primary endpoints are the number of hospital referrals and the diagnostic accuracy of the decision rule 24 hours, 6 weeks and 6 months after inclusion.

**Ethics and dissemination** The medical ethics committee Leiden-Den Haag-Delft (the Netherlands) has approved this trial. Written informed consent will be obtained from all participating patients. The results of this trial will be disseminated in one main paper and additional papers on secondary endpoints and subgroup analyses.

**Trial registration numbers** NL9525 and NCT05827237.

## STRENGTHS AND LIMITATIONS OF THIS STUDY

⇒ The study design is a randomised controlled diagnostic trial, which prospectively evaluates a clinical decision rule, including a high-sensitive cardiac troponin I (hs-cTnI) point-of-care test, to rule out acute coronary syndrome (ACS) in primary care.
⇒ The study includes a cost-effectiveness analysis, to evaluate the efficiency of the clinical decision rule and the impact on the healthcare system.
⇒ The Atellica VTLi hs-cTnI analyser provides reliable hs-cTnI measurements within only 8 min in capillary blood obtained by a fingerstick.
⇒ Inclusion is based on clinical suspicion of ACS by general practitioners, which reflects daily clinical practice, but may lead to inclusion bias.
⇒ Patients with an onset of chest pain <1 hour are excluded, because hs-cTnI measurement within this time window may be false negative due to time-dependent troponin release.

## INTRODUCTION

Chest pain is a common reason for consultation in primary care.[1 2] In 4%–7% of patients presenting with chest pain, the pain is caused by acute coronary syndrome (ACS).[1–3] Early identification and treatment of patients with ACS are important to avoid cardiac morbidity and mortality.[4] However, it is challenging for general practitioners (GPs) to distinguish ACS from other—less acute and life-threatening—causes of chest pain, as they have to rely on symptoms and clinical signs only.[5] Consequently, GPs refer approximately 40%–70% of patients presenting with chest pain to the hospital for further examination.[1 6]

Following referral, only 10%–20% of the patients are diagnosed with ACS.[7–9] The relatively high number of referrals puts a high burden on patients and may lead to

potentially avoidable adverse patient outcomes due to the risks associated with (unnecessary) diagnostic tests and overcrowding of emergency departments (ED).[10–12] In addition, a high referral rate puts a lot of pressure on the healthcare system. Finally, evaluation of patients in the ED is costly, which has a significant impact on society as well.[13] Therefore, measures to improve the referral efficiency of patients with acute chest pain are crucial.

A clinical decision rule may help GPs to safely rule out ACS and reduce hospital referrals. Previous studies in ED settings showed that a single high-sensitive cardiac troponin-I (hs-cTnI) measurement with or without a risk score is safe and effective to rule out ACS.[7 9 14] However, due to the low incidence of ACS in primary care, results from ED settings are not generalisable to primary care settings. Although use of a troponin point of care test (POCT) in primary care reduced the number of referrals in both a model-based and an observational study, the impact has yet to be studied in clinical practice.[15–17]

Therefore, this clustered randomised controlled trial (RCT) aims to evaluate the impact of a primary care decision rule for acute chest pain, consisting of a risk stratification score combined with a hs-cTnI-POCT on the number of hospital referrals, and to assess its diagnostic accuracy. In addition, secondary endpoints including cost-effectiveness will be assessed.

## METHODS AND ANALYSIS
### Study design
The POB HELP study is a clustered diagnostic RCT. The study will be conducted in general practices in three regions in the Netherlands (Leiden, Maastricht and Venlo). General practices will be randomised to the intervention or control group in a 2:1 ratio. Randomisation is stratified for the three regions to ensure even distribution of rural and urban areas. Randomisation is done by Castor Electronic Data Capture using a variable block randomisation method.[18] Patient inclusion started on 18 August 2021 and the study is planned to end on 1 July 2024.

### Study population
Patients visiting their GP, or visited by their GP, for acute chest pain symptoms in which the GP suspects an ACS, are eligible for inclusion. Patients who contact their GP by telephone with acute chest pain highly suspicious for ACS are not eligible for inclusion. For these patients, an ambulance is sent directly to the home address by the doctor's assistant, in accordance with usual care. Exclusion criteria are: age <18 years, trauma preceding the pain, inability to speak Dutch or understand the informed consent form, onset of chest pain within less than 1 hour and haemodynamic instability (table 1). Patients who match the eligibility criteria and provide informed consent are included in the study.

### Regular care
In general practices randomised to the control group, included patients will receive standard care in accordance

**Table 1** Inclusion and exclusion criteria

| Inclusion | Exclusion |
|---|---|
| ≥18 years of age | <1 hour since onset of symptoms |
| Acute chest pain | Inability to speak or understand Dutch |
| Seen by the general practitioner | Haemodynamic instability |
| | Trauma preceding chest pain |

with the Dutch GP guideline for ACS.[19] Standard care consists of a medical history and physical examination by the GP. In addition, an ECG can be performed, however, an ECG device is not available in many general practices in the Netherlands. Based on this information, the GP will decide to either reassure the patient ('watchful waiting'), consult a cardiologist or refer the patient to the hospital. The GP is responsible for the follow-up of the patient.

### Intervention
For patients included in general practices randomised to the intervention group, the GP will apply a clinical decision rule consisting of the Marburg Heart Score (MHS) and a hs-cTnI-POCT (figure 1).[3 20 21]

The MHS consists of a five-item score and each positive item results in 1 point: (1) age and sex (male ≥55 years, female ≥65 years), (2) history of cardiovascular disease (ie, stable coronary artery disease, prior myocardial infarction, coronary intervention, ischaemic cerebrovascular accident or peripheral atherosclerosis), (3) patient assumes a cardiac origin of the chest pain, (4) pain worsening with exercise and (5) pain not reproducible with palpation. The ACS risk is considered low for patients with an MHS of ≤2 points and high for patients with ≥3 points.[3 20]

The hs-cTnI measurement is performed using the Siemens Atellica VTLi immunoassay analyser.[21] The analyser is easy to use and guides the user step by step through the process. Prior to the start of the study, the staff, including the doctor's assistants and at least one GP per general practice, is trained by a study employee. A study employee is available during office hours to provide assistance by telephone when necessary. Blood is obtained by a fingerstick, which makes it highly suitable for a prehospital setting. The droplet of blood is applied to a cartridge using a capillary transfer device. The turn-around time from the application of the droplet of blood to the test result is approximately 8 min. The limit of detection is 1.6 ng/L and the limit of quantitation is 8.9 ng/L (at 10% CV) and 3.7 ng/L (at 20% CV).[22] The 99th percentile overall, for men and for women are, respectively, 23, 27 and 18 ng/L. A hs-cTnI level ≤3.8 ng/L is considered very low and is used as a cut-off in this study to exclude ACS.[23]

The combination of the MHS and the hs-cTnI-POCT measurement results in three possible recommendations (figure 1):

van den Bulk S, *et al. BMJ Open* 2023;**13**:e071822. doi:10.1136/bmjopen-2023-071822

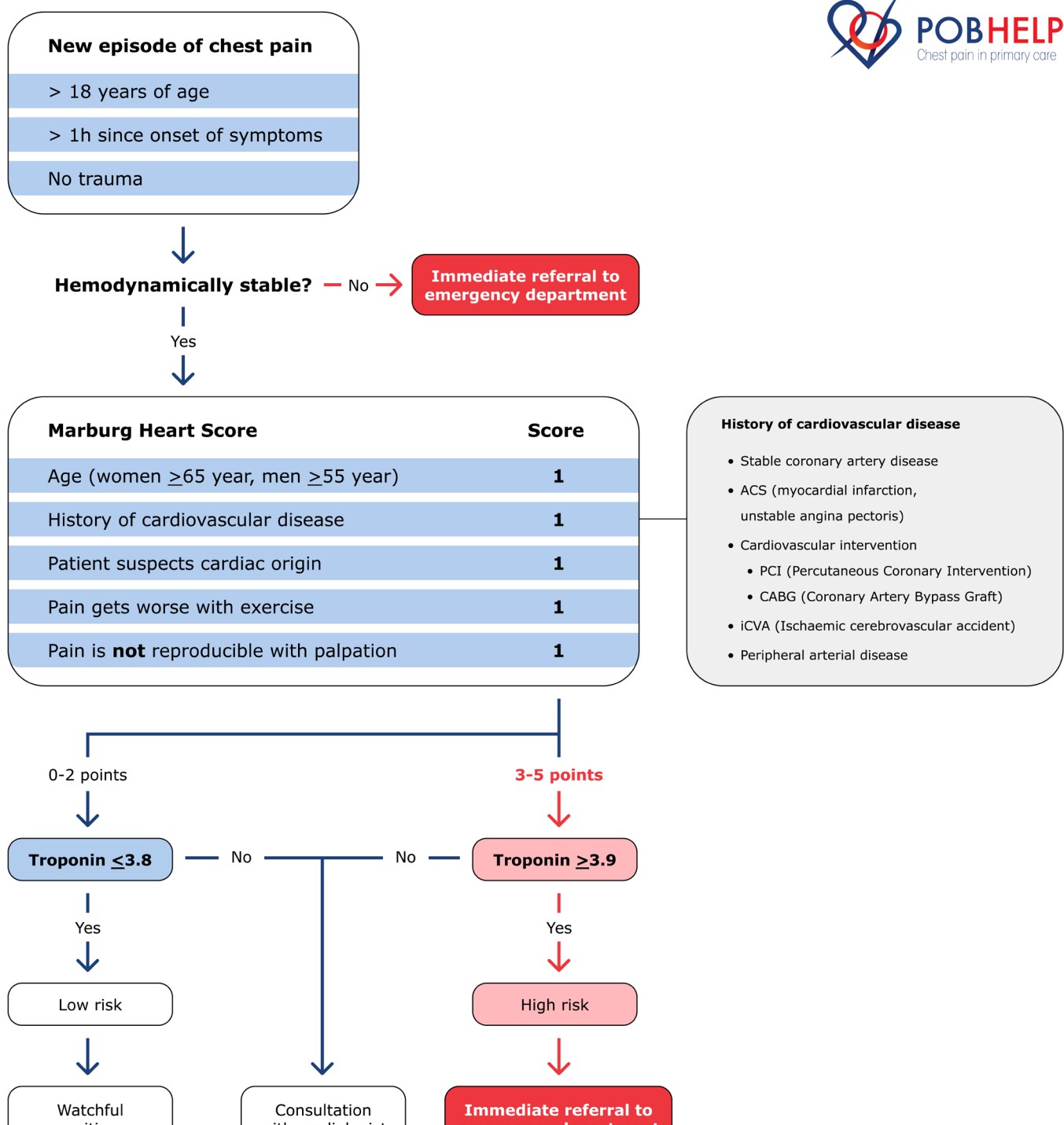

**Figure 1** Flow chart of the clinical decision rule.

1. MHS ≤2 points and hs-cTnI-POCT ≤3.8 ng/L: ACS is safely excluded and watchful waiting is recommended.
2. either MHS ≤2 points and hs-cTnI-POCT ≥3.9 ng/L or MHS ≥3 and hs-cTnI-POCT ≤3.8 ng/L: ACS cannot be excluded, consultation with a cardiologist is recommended.
3. MHS ≥3 and hs-cTnI-POCT ≥ 3.9 ng/L: The risk for ACS is increased and direct referral to the ED is warranted.

If the GP or consulting cardiologist disagrees with the recommendation provided by the clinical decision rule, they are allowed to overrule it. The GP is responsible for the patient's follow-up, in line with regular care.

In previous studies, the MHS showed good diagnostic accuracy in ruling out stable coronary artery disease in primary care, but was insufficient to rule out ACS safely.[3 8 20 24] We hypothesise that adding the hs-cTnI-POCT to the MHS will result in a safe and accurate clinical decision rule to rule out ACS in primary care. Even though the aim of the clinical decision rule is to rule out ACS in patients with acute chest pain, it may also reduce missed ACS diagnosis in patients presenting with atypical symptoms.

An ECG is not included in the decision rule, as an ECG device is not available in all general practices in The Netherlands. Nonetheless, GPs with an ECG device available are free to make an ECG and overrule the recommendations of the clinical decision rule as they deem necessary. However, it is important to notice that an ECG may be normal in 30% of patients presenting with non-ST elevated myocardial infarction and is, therefore, unsuitable for excluding ACS safely.[25]

### Safety

Following ESC guidelines the minimum time since onset of chest pain in this study is 1 hour.[25] Approximately 10% of all patients with acute chest pain in primary care present within 1 hour since onset of symptoms.[8 26] Due to the time-depended release of troponin, the decision rule is not applicable for these very early presenters.[25] Hs-cTnI measurements within this time window may give false negative results, and therefore, wrongfully rule out ACS. Previous studies including early presenters (<3 hours since onset of symptoms), showed decreased sensitivity in patients presenting <1 hour since onset of chest pain.[16 25–28] In patients presenting 1–3 hours since onset of chest pain, study results are still inconclusive, but using a hs-cTnI—with a cut-off value around the limit of detection, rather than below the 99th percentile—seems safe in combination with clinical evaluation and is recommended in the ESC guideline.[25 29 30] To ensure safety and minimise the risk of wrongfully excluding ACS, we took the following additional actions. First, patients contacting their GP by telephone with symptoms highly suspicious for ACS are not eligible for inclusion and an ambulance is sent to their home immediately. Patients who are haemodynamically unstable when seen by their GP are excluded as well and require immediate referral. Second, the hs-cTnI is embedded in a clinical decision rule, so patients with typical symptoms will be discussed with a cardiologist, even when hs-cTnI measurement is very low. Third, GPs are instructed that the recommendation of the clinical decision rule may be over-ruled when deemed necessary by the GP or consulting cardiologist. Lastly, a data safety monitoring board (DSMB) is installed to evaluate the safety of the study after 100, 500 and 1000 inclusions.

### Primary endpoints

The study has two primary endpoints. The first primary endpoint is the hospital referral rate for acute chest pain within 24 hours and 6 weeks after inclusion. The second primary endpoint is the diagnostic accuracy of the clinical decision rule. To study the diagnostic accuracy (ie, sensitivity, negative predictive value), the incidence of ACS and major adverse cardiac events (MACE) at the index consultation, within 6 weeks and 6 months after inclusion will be assessed. MACE is defined as a combined endpoint of ACS, percutaneous coronary intervention, coronary artery bypass grafting, coronary angiography revealing procedurally correctable stenosis managed conservatively and all-cause mortality.[7 31 32]

### Final diagnosis

The final diagnosis will be made based on GP's medical records, including all letters from hospitals. All cases will be reviewed by an independent endpoint committee consisting of a cardiologist and GP for adjudication of the final diagnosis and the occurrence of MACE. The committee will be blinded for the randomisation and hs-cTnI on presentation. In case of disagreement, a second cardiologist will be consulted. The final diagnosis will be classified as ACS, stable coronary artery disease, cardiac but non-coronary disease, non-cardiac chest pain or death. ACS comprises myocardial infarction (ST-segment elevation myocardial infarction (STEMI) and non-STEMI (NSTEMI)) and unstable angina pectoris. Myocardial infarction is defined conform the fourth universal definition of myocardial infarction.[33]

A coronary angiogram (CAG) is the golden standard to establish the presence of coronary artery disease. However, CAG is costly and not without risk (eg, bleeding in approximately 7% and procedure-related myocardial infarction in approximately 6% of patients undergoing CAG).[34 35] Therefore, we consider it too invasive and essentially unethical to perform a CAG in each patient in this relatively low-risk population. Hence, we will use a delayed reference standard, defined as ACS and MACE at the index consultation, within 6 weeks and 6 months after inclusion, to determine the diagnostic accuracy of the decision rule. The delayed reference method has been reported as a valid alternative if a golden standard test is too invasive and the disease is not self-limiting.[36] This method has been used in similar studies before.[7 20 32 37]

### Secondary endpoints

Secondary endpoints are the cost-effectiveness of the clinical decision rule, adherence to the recommendations of the clinical decision rule by GPs, the patient's reassurance, the diagnostic accuracy of the GP's gut feeling and the diagnostic accuracy of the HEART (History, ECG, Age, Risk factors and Troponin) score for all patients with an ECG available (box 1).[31] Finally, subgroup analyses for the primary endpoints will be performed for sex, region, socioeconomic status and duration of symptoms.

The cost-effectiveness of the clinical decision rule will be assessed alongside the trial. The analysis will be performed from a societal perspective. The costs are divided into healthcare costs and non-healthcare costs.

1. Cost-effectiveness, consisting of a trial-based cost analysis and a cost–utility analysis (costs per quality-adjusted life year (QALY), assessed using the EQ-5D-5L questionnaire).[39 40]
2. Adherence to the recommendations of the clinical decision rule by the general practitioner (GP), by comparing the GP's policy with the recommendations of the decision rule.
3. Reassurance of patients, using the State-Trait Anxiety Inventory after the index consultation.[41]
4. Diagnostic accuracy of the GP's gut feeling, by comparing the result of the gut feeling questionnaire with the occurrence of major adverse cardiac event (MACE) within 6 weeks after the index consultation.[42 43]
5. Diagnostic accuracy of the HEART (History, ECG, Age, Risk factors and Troponin) score for all patients with an ECG available for the occurrence of MACE within 6 weeks after the index consultation.[31]
6. Subgroup analysis for the primary outcomes for sex, region, socioeconomic status (using postal code) and duration of symptoms.

Healthcare costs include those of the intervention and other healthcare use during follow-up. The costs of the clinical decision rule will be based on microcosting including the duration of the GP consultation, medical equipment and materials used. Healthcare use will be assessed by questionnaires at 6 weeks and 6 months follow-up. In this questionnaires, patients will be asked to fill out their total healthcare utilisation, such as GP visits, outpatient clinic visits, ED visits and hospital admissions for any reason. Reference prices published in the Dutch costing manual will be used to value healthcare use.[38] To calculate non-healthcare costs, patients will be asked to note absence from paid and unpaid work. Productivity costs will be calculated using the friction cost method. Quality-adjusted life years (QALYs) will be calculated from utility scores from the EQ-5D-5L questionnaire administered at baseline, 6 weeks and 6 months follow-up.[39 40] Finally, costs and QALYs between the control and intervention groups will be compared.

To assess the GP's adherence to the recommendations of the decision rule, the outcome of the decision rule will be compared with the GP's policy as reported by the GP. To assess the effect of the clinical decision rule on patients' reassurance, patients will be asked to fill out the State-Trait Anxiety Inventory after the index consultation.[41] To evaluate the diagnostic accuracy of the GP's gut feeling, GPs will be asked to estimate the likelihood of ACS using the gut feeling questionnaire directly after the index consultation.[42 43] The outcome of this questionnaire will be compared with the occurrence of MACE within 6 weeks. In all patients with an ECG, the HEART score will be scored retrospectively based on the information on the inclusion form filled out by the GP. The outcome will be compared with the occurrence of MACE within 6 weeks.[31]

## Sample size calculation
In primary care in the Netherlands, the reported hospital referral rate of patients with chest pain ranges from approximately 40%–70%.[1 6] Similarly, a Swiss and Swedish study found referral and diagnostic testing rates of 59% and 43%, respectively.[44 45] Based on these referral rates, we conservatively assumed a hospital referral rate of 40%. For the primary endpoint, we estimated a 10% decrease in referrals, resulting in 30% referrals, which is in line with a previous study estimating a 10% decrease in referrals when using the MHS.[45 46] Group sample sizes of 448 (control group) and 896 (intervention group) achieve 80% power to detect the difference of 10% between both groups with a significance level (alpha) of 0.05 using a two-sided Z-test.[47] The sample size is adjusted for a clustering effect with an intraclass correlation coefficient of 0.05 and for 1:2 randomisation.[48] The estimated sample size needed is 500 for the control group and 1000 for the intervention group, accounting for a 10% lost to follow-up.

## Statistical analysis
Data will be analysed according to the intention-to-treat principle. Continuous demographic and baseline characteristics will be expressed as mean±SD and compared using the unpaired Student's t-test. Categorical demographic and baseline characteristics will be described as frequencies and percentages and compared using the $\chi^2$ or Fisher's exact test. The number of referrals will be presented as percentages and compared using the $\chi^2$ or Fisher's exact test. To assess the diagnostic accuracy of the decision rule, sensitivity, specificity and predictive values will be calculated. The overall diagnostic accuracy will be assessed by means of the C-statistics. Missingness will be assessed, and if the percentage of missingness is >10%, multiple imputation will be applied. All analyses will be corrected for the correlation within clusters (general practices), using generalised estimating equations. Subgroup analyses will be performed for sex, region, socioeconomic status and duration of symptoms to confirm the applicability of the decision rule for different subgroups.

## Clinical implications
In the Netherlands, GPs are the 'gatekeeper' to the hospital. Therefore, a clinical decision rule for acute chest pain is probably most beneficial in primary care. Based on previous literature, the clinical decision rule presented in this study may decrease hospital referrals of patients with acute chest pain by 10%.[45 46] Thereby, this strategy may reduce uncertainty for patients and doctors, increase patient satisfaction, decrease (unnecessary) diagnostic tests and relieve pressure on healthcare personnel and the healthcare system. Furthermore, it may lead to a substantial reduction in societal costs as was shown in a previous observational Norwegian study.[17] In the Netherlands, a patient referred to the hospital for acute chest pain on average costs €1580, whereas a primary care consultation plus hs-cTnI-POCT measurement is estimated to cost on average €73.[13 38] Therefore, if the decision rule would be implemented in 50% of general practices and assuming a 10% reduction in referrals, this could potentially lead to

an estimated cost reduction of €31 million in The Netherlands each year.[49]

## Generalisability

The current RCT will compare the clinical decision rule to regular care in general practices in three regions in The Netherlands, including urban and rural areas. After careful instruction on the inclusion criteria, the GPs in these practices will decide on the eligibility of individual patients. Although differences among GPs (eg, a less experienced GP may include different patients than an experienced GP) may lead to inclusion bias, this situation resembles daily clinical practice. Hence, the results of this study will be generalisable to the rest of The Netherlands and other countries with a similar primary care system, where the GP functions as a gatekeeper (eg, the UK, Scandinavian countries and Canada). In addition, we will measure the GP's working experience, which enables us to quantify this potential bias.

## Patient involvement

During the design of the study, the writing of the protocol and the development of patient information materials, a participant of 'Harteraad', a patient advisory council for heart disease, was involved.

## Ethics and dissemination

The study will be conducted following the Declaration of Helsinki. The medical ethics committee Leiden-Den Haag-Delft has approved the protocol. Leiden University Medical Center is the sponsor of the study. Written informed consent will be obtained from all participating patients. The trial is registered in the Netherlands Trial Register (NTR) (https://www.trialregister.nl/), NL9525. Accepted on 20 June 2021. Due to the current process of merging the NTR into a new national registry, changes after protocol amendments cannot be made. Therefore, we registered the trial a second time on https://www.clinicaltrials.gov/, NCT05827237. A DSMB has been installed evaluating the safety of the study after 100, 500 and 1000 inclusions.

Results of the study will be shared with professional healthcare workers, participants and the general public. The primary outcomes of the study will be presented in one main paper in a scientific peer-reviewed journal. The cost-effectiveness and other secondary outcomes will be shared in separate papers. Findings will be shared at conferences through oral and poster presentations. At the end of the study, all participants will be informed of the study results through email or on paper. End results and interim updates will be shared on the website: www.pobhelp.nl. Participating GPs and other stakeholders will be kept up to date through a 3 monthly digital newsletter.

## Author affiliations
[1]Public Health and Primary Care, Leiden Universitair Medisch Centrum, Leiden, The Netherlands
[2]Department of Family Medicine, Maastricht University, Maastricht, The Netherlands
[3]Cardiology, Leiden Universitair Medisch Centrum, Leiden, The Netherlands
[4]Cardiology, VieCuri Medisch Centrum voor Noord-Limburg, Venlo, The Netherlands
[5]Biomedical Data Sciences, Leiden University Medical Center, Leiden, The Netherlands

**Contributors** SvdB, TNB, RTAW, MEN and G-JD have written the protocol of the study. MJB, JGM and BMR substantially revised the protocol. MEvdA-vM has written the section on cost-effectiveness analyses. SvdB and AHJP have written this manuscript. All authors read and substantially revised the manuscript. All authors gave final approval of the published version.

**Funding** This work is supported by ZonMw (project ID 852001907). The Atellica VTLi immunoassay analysers used for the hs-cTnI-POCT measurements in this study are provided by Siemens Healthineers.

**Competing interests** None declared.

**Patient and public involvement** Patients and/or the public were involved in the design, or conduct, or reporting, or dissemination plans of this research. Refer to the Methods section for further details.

**Patient consent for publication** Not applicable.

**Provenance and peer review** Not commissioned; externally peer reviewed.

**ORCID iDs**
Simone van den Bulk http://orcid.org/0000-0002-0444-8932
Annelieke H J Petrus http://orcid.org/0000-0001-5967-805X
Robert T A Willemsen http://orcid.org/0000-0002-3538-0264
Mark J Boogers http://orcid.org/0000-0002-8755-4254
Joan G Meeder http://orcid.org/0000-0003-3261-9361
Braim M Rahel http://orcid.org/0000-0001-9048-1044
M Elske van den Akker-van Marle http://orcid.org/0000-0002-5269-509X
Mattijs E Numans http://orcid.org/0000-0002-0368-5426
Geert-Jan Dinant http://orcid.org/0000-0001-7411-8325
Tobias N Bonten http://orcid.org/0000-0002-7719-6182

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
