## [Reviewer comments · BMJ Open]

ARTICLE DETAILS

TITLE (PROVISIONAL)	Ruling out acute coronary syndrome in primary care with a clinical decision rule and a capillary, high-sensitive troponin I point of care test: Study protocol of a diagnostic RCT in the Netherlands (POB HELP)
AUTHORS	van den Bulk, Simone; Petrus, Annelieke; Willemsen, Robert; Boogers, Mark; Meeder, Joan; Rahel, Braim; van den Akker-van Marle, Elske; Numans, Mattijs; Dinant, Geert-Jan; Bonten, Tobias N.

VERSION 1 – REVIEW

REVIEWER	Paul Collinson St Georges Hospital, Clinical Blood Sciences
REVIEW RETURNED	25-Jan-2023

GENERAL COMMENTS	The authors propose to undertake a fascinating study of the potential for utilising point of care testing (POCT) in primary care to measure high sensitivity cardiac troponin I (hs cTnI) as a means of primary care based triage to avoid hospital admission. This has the goal of reducing the number of referrals to the Emergency Department (ED). This is a goal which will unite physicians, hospital administrators and politicians. In terms of the study I have the following observations to make which the authors may wish to consider. 1. Patient selection. The exclusion criteria include patients with pain of <1 hour duration. The current European Society of Cardiology guidelines do endorse the use of a one hour cut off for chest pain but this recommendation has been criticised as there is relatively little evidence when chest pain is of <3 hour duration. In a recent study (currently submitted for publication) examining the hs cTnI method from Abbott specifically in patients presenting <2 hours we have found that a lower cut off may be appropriate (<2 ng/L rather than <5 ng/L). These factors are also discussed in Crea F, Jaffe AS, Collinson PO, Hamm CW, Lindahl B, Mills NL et al. Should the 1h algorithm for rule in and rule out of acute myocardial infarction be used universally? Eur Heart J 2016;37:3316-23. Apple FS, Collinson PO, Kavsak PA, Body R, Ordonez-Llanos J, Saenger AK et al. Getting Cardiac Troponin Right: Appraisal of the 2020 European Society of Cardiology Guidelines for the Management of Acute Coronary Syndromes in Patients Presenting without Persistent ST-Segment Elevation by the International Federation of Clinical Chemistry and Laboratory Medicine Committee on Clinical Applications of Cardiac Bio-Markers. Clin Chem 2021;67:730-5.
--

	2. Sample timing and selected cut off. Following on from the point above the selected cut off for the POCT method comes from the observational study of Apple et al. Examination of the published data shows that > 70% of the presentations were > 2 hours and median time to first sample varied from 34 to 54 minutes. The implication is that the majority of patients were probably sampled more than three hours from onset of chest pain. This means that the optimised cut-off used may be slightly too high and not be appropriately representative for the proposed study population studied. Unfortunately, the authors do not indicate the likely time of presentation in primary care so it is difficult to judge whether this is likely to have an impact. Both of the points above means that there may not be a detectable rise in troponin from ischaemic myocardial injury measurable in the population undergoing investigation. This does not necessarily mean that the methodology does not work. The baseline level of troponin is an excellent predictor of future cardiovascular events (for hs cTnI) so this may itself be additive to a clinical risk score. In addition, the negative predictive value of a test strongly influenced by disease prevalence. When disease prevalence is low even if the test sensitivity is relatively low, negative predictive value will be high. 3. In terms of sample size and sample size calculation, what is the prevalence of ACS in patients presenting to general practice so how many patients would an individual general practitioner expect to see. This will have an impact on the ability to assess patients likely as well as the ability to perform POCT. If the POCT instrument is only being used once weekly, the operator is likely to be less proficient than if it is being used on a daily basis. 4. Cost and implementation of POCT. The challenges of implementation of POCT into routine clinical practice should not be underestimated. For example, see. Abel, G., Brugnara, C., Dsa, S., David, K., Deaton-Mohney, E. B., Halverson, K., Mann, P. A., Moore, D., Nichols, J. H., Ondracek, C. R., and Korpi-Steiner, N. Point of Care Testing: A "How-To" Guide for the Non-Laboratorian. AACC https://www.aacc.org/pocthowto. 2022. These may also influence the costing of POCT. As discussed above, the number of patients seen by an individual general practitioner/practice will influence the number of instruments required (which have a capital cost) which should normally be a minimum of two per test site (to provide backup) as well as the quality assurance regime hence number of test cartridges used. Costing should also factor in training costs and cost of ongoing support.
--	---

REVIEWER	Oscar Pereira Dutra Instituto de Cardiologia do Rio Grande do Sul
REVIEW RETURNED	22-Feb-2023

GENERAL COMMENTS	this study flexibility in the care and possible diagnosis of chest pain
---

VERSION 1 – AUTHOR RESPONSE

Reviewer 1

We thank the reviewer for his enthusiastic view on our study and his fair and thoughtful remarks.

1. We have read the suggested articles with great interest and acknowledge the concerns raised regarding the early presenters. We would like to elaborate on our choice and the actions we have taken to minimise the risk of missing acute coronary syndrome in our study.

We have chosen to exclude patients presenting with chest pain <1 hour, following the recommendation of the current European Society of Cardiology guidelines. The guideline states the following:

'In patients with MI, levels of cardiac troponin rise rapidly (i.e. usually within 1 h from symptom onset if using high-sensitivity assays) after symptom onset and remain elevated for a variable period of time (usually several days).' (3.3.2 Biomarkers: high-sensitivity cardiac troponin) and:

'The ESC 0 h/1h and 0 h/2 h algorithms apply to all patients irrespective of chest pain onset. The safety (as quantified by the NPV) and sensitivity are very high (>99%), including in the subgroup of patients presenting very early (e.g. <2 h). However, due to the time dependency of troponin release and the only moderate number of patients presenting <1 h after chest pain onset in previous studies, obtaining an additional cardiac troponin concentration at 3 h in patients presenting <1 h and triaged towards rule-out should be considered.' (3.3.4.1 Caveats of using rapid algorithms – vii)

Crea et al. and Apple et al. mainly raise the concern that there is relatively little evidence in patients presenting early after symptom onset and question if it is save enough to use the 0h/1h-algorithm in early presenters. In the response of Mueller et al. a study is mentioned that validated this algorithm in 1282 patients with a median time from chest pain onset to presentation of 1.8h (interquartile range 1.0-2.9h) with 213 patients (16.6%) having myocardial infarction. Blood drawl was within 45 minutes. 7 patients with MI were missed. However, the hs-troponin cut off was 12ng/L, which was higher than the limit of detection. When using a lower cut-off closer to the limit of detection (for example 5ng/L or 3ng/L), only 3 or 0 patients would have been missed respectively.

Various studies focused on early presenters. For example

1. Johannessen et al included 1711 patients in a low-risk pre-hospital setting, validating the 0h/1h algorithm of whom 182 presented <3 hours since onset of symptoms. 1 out of the 1241 patients in the rule out group was falsely classified as low risk. However, this patient presented 18 hours after onset of symptoms.
2. Ljung et al evaluated 911 NSTEMI patients presenting <2h after symptom onset with 10-14 minutes from ED presentation to 1st troponin. In the group presenting 1-2h since symptom onset, 10/654 patients with NSTEMI presented with hs-troponin under the limit of detection. They found that a single hs-troponin alone is insufficient, but it can be used in patients presenting 1-2 hour since onset of chest pain in combination with an ECG (Sensitivity 99.4%).
3. Andersen et al included 470 early presenters (<3h) of which 140 patients presented <1h and 330 patients 1-3h since onset of chest pain. The median time to blood drawl was 51 minutes. No MI patients were missed in these early presenters. (Sensitivity 100%)
4. Body et al included 324 patients presenting <3h since onset of symptoms of whom 79 had a myocardial infarction. 1 patient presented with hs-troponin under the cut-off point and showed elevation later. This patient presented <1hour since onset of symptoms.
5. Sandoval et al included 2212 patients, of which 795 patients were classified as early presenters (<3h). They found no statistical differences in the safety of this subgroup.

These studies show that sensitivity in early presenters is high when using a very low cut-off point, as we do in our study. However, sensitivity is not 100% and the chance of false negative results seems

higher in the group presenting very early than in late presenters. Therefore, we took the following additional precautions:

1. We have validated the safety of our cut-off point of 3.9ng/L in blood samples of an early presenter cohort from the FAMOUS trial with patients presenting <2 hours and <1hour since onset of chest pain. (unpublished)
2. Patients with a very high suspicion of ACS are not eligible for inclusion. Doctor's assistants in the Netherlands are trained to directly send an ambulance to the home of patients calling with typical acute chest pain symptoms.
3. The single hs-troponin measurement is combined with a risk score, to make sure that presentations highly suggestive for ACS, are at least consulted with a cardiologist even when hs-troponin I measurement is very low.
4. General practitioners are instructed to overrule the recommendation of the decision rule as they deem necessary/ based on their gut feeling. Meaning they can still refer their patients to the ED, even when hs-troponin I is very low.
5. A data safety monitor board (DSMB) is installed to look specifically at the safety of the study. They will meet after 100, 500 and 1000 inclusions.

- We added point 2 (described above) to the method – study population section.

Page 5, line 93-96 and in table 1 In- and exclusion criteria, page 5, line 100

- We added a new 'Safety' paragraph in the method section, in which we describe the additional safety measures just mentioned.

Page 8-9, line 169-189

2. The concern raised by the reviewer is that the time between presentation and first blood draw in studies used to determine the cut-off is 34 to 54 minutes. Meaning that patients classified as early presenters (i.e. presenting <3h) do not automatically have their first hs-troponin measurement within this time window.

Time of presentation in primary care varies. A Dutch study by Schols et al. found that 9.5% of patients presented <1 hour since onset of symptoms, 56.8% between 1-24h since onset of symptoms and 33.7% >24h since onset. The OUT-ACS study in Norway found similar percentages: 10.6% <3h since onset of symptoms, 35.6% 3-6h since onset and 23.9% 6-12h since onset.³

We looked at the first 200 inclusions of our study and found a similar distribution for the time since symptom onset. With 11% being early presenters (<3 hours since onset of symptoms) and the majority having chest pain for more than 6 hours (77%).

We estimate that the time from presentation to blood sampling will be 15-30 minutes, due to the initial examination by the general practitioner, informed consent procedure and preparation by the assistant.

This means that only a small proportion of the patients included will be early presenters, in a low risk population. Adding the additional safety measures that we installed as mentioned in the response to question 1, we expect the clinical decision rule to be safe.

We added a paragraph in the method section 'Safety' Page 8-9, line 169-189, in which we mention the additional safety measures that we installed.

3. Two Dutch studies included patients with chest pain suspected of ACS in general practice during 2 weeks.^{8,9} On average 1-2 patients per practice were included during this 2 weeks.

We agree that the frequency of performing the POCT test may contribute to the proficiency. However, the analyser used is very user friendly and guides the user step by step through the process. It has many safety checks preferring no result over a wrong result. Furthermore, a study assistant is

available during the day to assist over the phone. Lastly, doctor's assistants are used to POCT equipment which is not used on a daily basis in primary care. For example, CRP and D-dimer measurements. Therefore, we do not expect the proficiency of the doctor's assistant to influence the troponin value and therefore patient safety.

We added information on the user friendliness of the analyser in the method section -'intervention'.
Page 6, line 126-130

4. We fully agree with the reviewer that implementation will be the next big challenge after finishing this trial. Experiences gained during this trial will provide valuable information on potential facilitators and barriers. However, the first step is to complete the trial to establish the safety and efficiency of the clinical decision rule including the hs-troponin measurement in primary care. During the study we also collect data on the number of tests performed and the number of tests failed. We can use this information to determine how much instruments are needed on average per patient and if and how this changes with increased experience. We will take this into account in our cost analyses, together with quality controls, training costs and technical support.

Reviewer 2

We thank the reviewer for reviewing our manuscript. Unfortunately, we are not sure what the reviewer means by his comment. If the reviewer wishes us to change something in the manuscript or to respond to the comment, we kindly ask for an additional explanation.

Additionally we made some textual changes, which we believe add to the readability of the paper.

VERSION 2 – REVIEW

REVIEWER	Paul Collinson St Georges Hospital, Clinical Blood Sciences
REVIEW RETURNED	10-May-2023
GENERAL COMMENTS	Thank you for the excellent responses to my comments.